# Study on the Hemostasis Characteristics of Biomaterial Frustules Obtained from Diatom *Navicula australoshetlandica* sp.

**DOI:** 10.3390/ma14133752

**Published:** 2021-07-05

**Authors:** Yanqing Luo, Shuangfei Li, Kun Shen, Yingjie Song, Jiangtao Zhang, Wen Su, Xuewei Yang

**Affiliations:** 1Guangdong Technology Research Center for Marine Algal Bioengineering, Guangdong Key Laboratory of Plant Epigenetics, College of Life Sciences and Oceanography, Shenzhen University, Shenzhen 518060, China; lyanqingl87@163.com (Y.L.); sfli@szu.edu.cn (S.L.); 2018304053@email.szu.edu.cn (K.S.); yjsong0517@163.com (Y.S.); 2Shenzhen Key Laboratory of Marine Biological Resources and Ecology Environment, Shenzhen Key Laboratory of Microbial Genetic Engineering, College of Life Sciences and Oceanography, Shenzhen University, Shenzhen 518055, China; 3Longhua Innovation Institute for Biotechnology, Shenzhen University, Shenzhen 518060, China; 4Shenzhen Jawkai Bioengineering R & D Center Co., Ltd., Shenzhen 518120, China; zhangzzjt@163.com; 5Department of Pathology, Shenzhen University Health Science Center, Shenzhen 518055, China; suwen@szu.edu.cn

**Keywords:** *Navicula australoshetlandica* sp., biosilica frustule, mesoporous biomaterial, hemocompatibility, hemostasis characteristics

## Abstract

Diatoms, known as photosynthetic unicellular algae, can produce natural biosilica frustules that exhibit great biocompatibility, superhydrophilicity, and superhemophilicity. In our study, the diatom *Navicula australoshetlandica* sp. was isolated from aquaculture wastewater and pretreated to obtain frustules so as to explore their hemostasis characteristics. A special “porous web” (6–8 nm) substructure in the ordered nanopores (165–350 nm) of boat-shaped diatom frustule was observed in *Navicula australoshetlandica* sp. using SEM and TEM analysis. Moreover, X-ray, N_2_ adsorption–desorption isotherms, and BET analysis showed that the diatom frustule is a mesoporous material with a surface area of 401.45 m^2^ g^−1^ amorphous silica. FTIR analysis showed that *Navicula australoshetlandica* sp. frustules possessed abundant OH functional groups. A low hemolysis ratio was observed for 1–5 mg mL^−1^ diatom frustules that did not exceed 1.55 ± 0.06%, which indicates favorable hemocompatibility. The diatom frustules exhibited the shortest clotting time (134.99 ± 7.00 s) with a hemostasis material/blood (mg/μL) ratio of 1:100, which is 1.83 times (112.32 s) shorter than that of chitosan. The activated partial thromboplastin time (aPTT) of diatom frustule was also 44.53 s shorter than the control. Our results demonstrate the potential of *Navicula australoshetlandica* sp. diatom frustules to be used as medical hemostasis material.

## 1. Introduction

Diatoms, known as photosynthetic unicellular algae, can utilize sunlight energy to synthesize organic compounds in both marine and fresh water environments [1]. Moreover, diatom cultivation can be applied as a sustainable solution for removing the excessive nitrogen and phosphorus nutrients in aquaculture wastewater [2]. Meanwhile, after removing the organic substance, the remaining silica skeletons become considerably useful as diatom frustules. This is due to their unique porous cell walls possessing species-specific shape and organized pore arrangement [3]. With the advantages of excellent biocompatibility [4], hydrophilic surface, and being microsized to nanosized, diatom frustules have attracted great attention regarding their potential application as nanomaterials in a medical context. By simply treating them with acid or high-temperature baking [5,6], we can obtain mass porous biosilica without toxic reagents or laborious and complicated synthesis processes. Diatoms are a promising resource for providing rigid and complex 3D biosilica structures in the microscale to nanoscale range, with their large surface area (~100 m^2^ g^−1^), mechanical resistance, and suitability for easy chemical modification [7]. The above characteristics present a compelling argument for the application of diatom biosilica as a medical nanomaterial for drug delivery, for biosensors, and as a hemostatic agent [8,9,10,11]. Moreover, studies have been carried out demonstrating the prominent hemostatic ability of mesoporous silica spheres [12,13].

However, the synthesis of mesoporous silica requires multiple agents, co-solvents, or additives as well as the finer control of the synthesis process to create the desired shape and size [14,15]. Thus, biosilica frustules, which originate from a natural aquatic plant and exhibit attractive features of biocompatibility, superhydrophilicity, and superhemophilicity, have been considered in greater detail [16]. The effective performance of diatom (*Coscinodiscus* sp.) biosilica in halting bleeding was demonstrated by Feng et al. in 2016. They conducted a rat-tail amputation experiment and found that the clotting time of diatom (160.78 ± 28.56 s) was much shorter than that of gauze (510.26 ± 63.22 s) [17]. In 2020, Jeehee et al. used diatom silica prepared from *Melosira nummuloidese* and showed that the blood contact angle of the diatom frustule (69.55 ± 8.95° at t = 0 s) was nearly 0° after 10 s, thus demonstrating the superhydrophilicity and superhemophilicity of the diatom biosilica. In 2019, Wang et al. investigated the homeostatic performance of *Thalassiosira weissflogii*, *Thalassiosira* sp., and *Cyclotella cryptica*, with *T. weissflogii* exhibiting the shortest hemostasis time (158 ± 8.19 s) for in vitro blood coagulation [18]. Research on the hemostatic ability of *Navicula* sp. conducted by Wang et al. showed a 14.6% shorter in vitro coagulation time than Quikclot [11] However, given the equivalent surface area, the particle diameter of *Navicula* was shorter than that of *Coscinodiscus*. Relevant research on nanoscale silicon frustules of *Navicula* species are still scarce. Furthermore, mass affiliation of *Navicula* under aquacultural wastewater can sharply reduce the costs of diatom frustule preparation.

In our study, we aimed to explore the potential of diatom *Navicula australoshetlandica* sp. frustules for hemostasis. We also sought to examine their physical and chemical characteristics using scanning electronic microscopy (SEM), transmission electron microscopy (TEM), Fourier-transform infrared spectroscopy (FTIR), X-ray diffraction (XRD), and Brunauer–Emmett–Teller (BET) analyses. The hemostatic efficiency (in vitro hemolysis test, in vitro blood clotting evaluation, activated partial thromboplastin time, and prothrombin time) of *Navicula australoshetlandica* sp. was also studied.

## 2. Materials and Methods

### 2.1. Diatom Species and Culture Conditions

Diatom species (*Navicula australoshetlandica* sp. L1) were screened from the golden pompano breeding base (22°35′16″ N, 114°28′47″ E) in Dapeng, Shenzhen, China. *Navicula australoshetlandica* sp. L1 was cultivated in f/2 medium-enriched seawater sodium with Na_2_SiO_3_ (0.04 g L^−1^) [19] at 30‰ salinity. A concentration of 100 mg L^−1^ streptomycin sulfate was added to avoid bacterial contamination. The seed inoculum of diatom *Navicula australoshetlandica* sp. was thus grown at 25 °C under aeration, with a photon flux density of 50 μmol photons m^−2^ s^−1^ 12 h illumination/12 h darkness cycle in an illumination incubator (MGC-450BP-2, bluepard instruments, Shanghai, China). A total of 300 mL f/2 medium in a 1000 mL Schott bottle were inoculated with 10 mL (10% (*v*/*v*) inoculation ratio) of the above culture. Three biological replicates of each sample were examined.

During the cultivation period, diatom density was determined daily at the stationary phase by measuring the optical density at 650 nm with cell counting. Before sampling, flasks were stirred for 2 min, allowing the dispersion of the deposited and aggregate diatom cells and nutrient homogenization. Diatom samples were counted using a 0.1 mL counting chamber (DSJ-01, Dengxun, Xiamen, China) and light microscopy (DMi1, Leica, Wetzlar, Germany). All experiments were conducted in triplicate.

### 2.2. Cultivation of Navicula australoshetlandica sp. in Artificial Aquaculture Wastewater

Artificial aquaculture wastewater was prepared as shown in Table 1, which lists the physicochemical parameters of the wastewater used for the experiments. Silicon was also added for diatom-specific needs. The diatom used to purify the AW was firstly cultivated in standard f/2 sodium with Na_2_SiO_3_ (0.04g L^−1^) for 7 days. Then, 100 mL diatom seed inoculum was centrifuged at 8000 rpm min^−1^ for 15 min and the precipitate was retained and washed three times with ddH_2_O. Subsequently, the diatom cells were transferred to 250 mL artificial aquaculture wastewater (AW) in a 500 mL flask for 14 day growth. The cultivate conditions were the same as in Section 2.1 (without aeration) and were conducted in triplicate.

### 2.3. Preparation of Diatom Frustules

Following the 14 day cultivation of diatom in artificial aquaculture, the diatom cells were collected by centrifugation at 8000 rpm min^−1^ for 15 min. Then, the diatom sediment was frozen at −80 °C for 12 h and subsequently placed in a vacuum-freeze drier for 24 h. After that, 200 g diatom powder was suspended in 15 mL piranha solution (sulfuric acid/H_2_O_2_ = 7:3) for a cleaning routine at 70 °C for 60 min. Then, the diatom frustules were collected by filtration with a polytetrafluoroethylene (PTFE) filter (50 mm diameter, 0.7 μm pore filter, Jinteng, Tian Jin, China) and rinsed 5 times with deionized water. Finally, frustules were dried using a vacuum-freeze drier (Triad 2.51, Labconco, Kansas City, MO, USA) for 24 h.

### 2.4. Physical and Chemical Characterization for Diatom Frustules

The morphology of diatom frustules was observed via scanning electron microscopy (SEM, APREO S, Thermo Fisher Scientific, Waltham, MA, USA) and transmission electron microscopy (TEM, HT7700, HITACH, Tokyo, Japan). Specimens were dispersed in ethanol before being dropped on 5 mm × 5 mm silicon wafers and formvar/carbon film and dried at room temperature. Specimens for SEM were sputtered with 10 nm gold using a high vacuum coater (EM-ACE600, Leica, Wetzlar, Germany). Simultaneously, the surface element was analyzed by energy dispersive X-ray spectroscopy (EDX). The chemical structures were characterized by Fourier transform infrared spectroscopy (FTIR, Nicolet 6700, Thermo Fisher Scientific, Waltham, MA, USA) using the KBr method with 5 mg diatom frustules in 100 mg of KBr powder. The FTIR spectra were collected over the range of 4000–400 cm^−1^ with a resolution of 4 cm^−1^ in transmission mode. Surface areas, pore volume, and pore width distribution were measured by nitrogen adsorption/desorption isotherms using a volumetric adsorption analyzer (ASAP 2020, Micromeritics, Norcross, GA, USA). The specific surface area was calculated using Brunauer–Emmett–Teller (BET) methods. Fifty milligrams diatom frustules are needed for the BET analysis compared with the 2 mg used for SEM observation. Nitrogen adsorption measurements were carried out at five different partial pressures (P/P_0_ 0.10–1), with a standard deviation between replicate measurements of <1%. Diatom crystallinity was performed using a powder X-ray diffractometer (XRD, PANalytical B.V., Almelo, The Netherlands) and by using CuKα radiation over the range of 10 to 80°.

### 2.5. In Vitro Hemolysis Test

The hemolysis ratio of diatom frustules was tested in vitro. A 10 mg mL^−1^ diatom frustule saline stock liquid was prepared and progressively diluted with saline at different concentrations (1.0, 2.5, 5.0, 7.5, and 10.0 mg/mL). Different concentrations of diatom frustules saline were pre-warmed to 37 °C. After that, 60 μL RBC dispersions (the precipitation after whole blood centrifuge at 3000 rpm min^−1^ for 15 min) were added into the frustule suspensions (1 mL) and incubated at 37 °C for 1 h. Distilled water and saline without frustule suspensions were used as positive control and negative control, respectively, and also incubated at 37 °C for 1 h. Then, the mixture compounds and controls were centrifuged at 2000 rpm min^−1^ for 5 min. A 100 μL aliquot of supernatant was taken for measurement at 545 nm using a microplate reader (EPOCH2, BioTek, Winooski, VT, USA). The hemolysis rate (HR, %) was calculated using Equation (1). The hemolysis ratio of diatom frustules was tested in vitro. A 10 mg mL^−1^ diatom frustule saline stock liquid was prepared and progressively diluted with saline at different concentrations (1.0, 2.5, 5.0, 7.5, and 10.0 mg/mL). Different concentrations of diatom frustules saline were pre-warmed to 37 °C. After that, 60 μL RBC dispersions (the precipitation after whole blood centrifuge at 3000 rpm min^−1^ for 15 min) were added into the frustule suspensions (1 mL), distilled water and saline without frustule suspensions (1 mL) and incubated at 37 °C for 1 h. Distilled water and saline without frustule suspensions were used as positive control and negative control, respectively. Then, the mixture compounds and controls were centrifuged at 2000 rpm min^−1^ for 5 min. A 100 μL aliquot of supernatant was taken for measurement at 545 nm using a microplate reader (EPOCH2, BioTek, Winooski, VT, USA). The hemolysis rate (HR, %) was calculated using Equation (1).
(1)$$Hemolysis\;ratio\left(\%\right)=\frac{D_{F}-D_{P}}{D_{N}-D_{P}}\times 100$$

In this equation, *D_F_*, *D_N_*, and *D_P_* are the absorbance at 545 nm of the frustule sample, distilled water (negative control), and saline (positive control), respectively.

### 2.6. In Vitro Blood Clotting Evaluation

Sprague Dawley rats at 7–8 weeks old (average in female and male) were purchased from Guangdong Medical Laboratory Animal Center, China. The experiment was approved by the Experimental Animal Ethics Committee (AEWC) of Shenzhen University with the Approval No. 2021002

#### 2.6.1. The Whole Blood Clotting Test

The whole blood clotting test was performed according to [20]. Both diatom frustules and chitosan were pretreated at 37 °C, respectively. Diatom frustules (5–30 mg) and chitosan (5–30 mg) were added along with 1 mL of anticoagulant to rat blood. Then, 0.2 M CaCl_2_ was added to the blood–sample mixture at a 1/50 V ratio. The re-calcified blood without diatom frustule and chitosan was applied as control. The clotting times were recorded as the point where blood completely ceased flowing. After the blood coagulation, the samples were rinsed with phosphate buffer solution (pH 7.4) three times to discard the physically adhered cells. Then, the samples were fixed with 2.5% glutaraldehyde at 4 °C for 2 h. The blood cells with diatom frustules were dehydrated with graded alcohol and dried at room temperature and were subsequently observed by SEM, according to Section 2.4.

#### 2.6.2. Blood Coagulation Tests

The activated partial thromboplastin time (aPTT) and prothrombin time (PT) were analyzed to check the coagulation tests. A semiautomatic coagulation analyzer (XN06-II, KING DIAGENOSTIC, Wuhan, China) was applied to perform the tests with blood taken from rats. The inferior vena cava was mixed with 3.8% sodium citrate (1/10 volume) and then centrifuged at 37 °C at 3000 rpm for 15 min. The blood serum (supernatant) was collected after the centrifuge. In order to analyze PT, 100 μL of PT reagent was firstly pre-warmed for 3 min at 37 °C. After the pretreatment, 50 μL of serum was mixed with 100 μL of the PT reagent. All samples were required to be preincubated at 37 °C for 3 min before being measured with a semiautomatic coagulation analyzer. In order to analyze aPTT, 50 μL of blood serum was mixed with 50 μL of aPTT reagent and incubated at 37 °C for 3 min. Then, 50 μL of 0.025 mol L^−1^ CaCl_2_ was added into 100 μL of mixture to analyze aPTT with a semiautomatic coagulation analyzer. All the experiments were conducted using six duplicates in each sample.

### 2.7. Analytical Methods

All the values are expressed as the mean ± standard deviation (SD). All the statistical analyses, including Student’s *t*-tests and ANOVA tests, were conducted using the statistical software GraphPad Prism 8. A value of *p* < 0.05 was considered to indicate statistically significant differences.

## 3. Results

### 3.1. Physical and Chemical Characteristics Analysis of Diatom Frustules

The symmetric *Navicula australoshetlandica* sp. diatom was isolated from aquaculture wastewater and cultivated in the conditions listed in Section 2.1. The diatom frustule is the silica scaffold of a diatom cell without organic residues. After the corrosion and oxidation of piranha resolution, the frustule still retained the original “boat” shape and nanoporous arrangement. Figure 1a shows the original shape and the color of the *Navicula australoshetlandica* sp. The apical axes of diatom cells varied from 8 to 11 μm, with the diameter of the highly ordered fracture porous pattern ranging 165–350 nm (Figure 1b). Figure 1c shows the web substructure of the nanoporous *Navicula australoshetlandica* sp. frustules revealed for the first time. It clearly shows that there is an ordered “porous web” (with a diameter of 6–8 nm) inside the nanoporous *Navicula australoshetlandica* sp. frustule. This web substructure could greatly increase the surface area of the *Navicula australoshetlandica* sp. frustule, which is beneficial in terms of its potential utilization as a mesoporous biomaterial.

The XRD patterns of *Navicula australoshetlandica* sp. diatom frustules are shown in Figure 2a. The broad peak centered at 23° (2θ) indicates the presence of amorphous silica. The Fourier transform infrared spectroscopy (FTIR) experiments further confirmed the abundant silanol groups in diatom silica frustules. The bare frustule surface (Figure 2b) showed one strong peak bonding at 1091 cm^−1^ that corresponds to asymmetric Si–O–Si stretching vibrations and the bands at a wavelength of 799 cm^−1^ originally belong to symmetric Si–O–Si stretching [21]. A silanol group Si–OH stretching vibration was detected at around 949 cm^−1^. Additionally, a broad peak centered at 3444 cm^−1^ and the peak at 1634 cm^−1^ can be separately attributed to OH stretching and the reflection of OH bending of water [22]. All the bands reflect the high abundance of OH, which provides the ultra-hydrophilic properties of frustules.

The Brunauer–Emmett–Teller (BET) method was applied in the analysis of the specific surface area (Table 1). Pore width distribution was estimated from the quantity adsorption branch of N_2_ isotherm using the Barrett–Joyner–Halenda (BJH) method (Table 2).

The N_2_ adsorption–desorption isotherms of the diatom frustules (Figure 2c) show a sharp adsorption step at P/P_0_ values between 0 and 0.1, with a small degree of hysteresis at P/P_0_ values between 0.8 and 0.99 corresponding to a type IV isotherm, which is typical of mesoporous materials [23]. From nitrogen adsorption–desorption measurements, diatom frustules have Brunauer–Emmett–Teller (BET) surface areas of 401.45 m^2^ g^−1^. The BJH pore diameter was 8.58 nm, which corresponds to the pore diameter distribution (Figure 2c) and TEM images (Figure 1c). Results show that diatom frustules of *Navicula australoshetlandica* sp. exhibit a multiscale hierarchical porous structure with sizes ranging from 10 to 100 nm. This demonstrated that the diatom frustules have high surface area and high porosity, where pore width is mainly around 4.61 nm.

### 3.2. In Vitro Hemolysis Test

A hemolysis test was introduced to confirm the compatibility of the hemostatic utilization of diatom frustules. This test gauges the hemolytic phenomenon mainly caused by electrostatic interactions between the silanol groups distributed on the surface of diatom frustules and the positively charged groups of membrane proteins. The strong affinity between silica and membrane weakens the integrity of red blood cells [24]. The hemolysis ratio of red blood cells (RBCs) basically turned higher when the concentration of diatom frustules was increased, as shown in Figure 3a. In order to quantify the hemolytic effects of diatom frustules in SD rats erythrocytes, hemolysis ratio was calculated (according to Equation (1)). From Figure 3b, a low hemolysis ratio was found in 1, 2.5, and 5 mg mL^−1^ diatom frustules, with no more than 1.55 ± 0.06%. The lowest hemolysis ratio was 1.18 ± 0.22% in 1 mg mL^−1^ diatom frustules, which can be recognized as a nearly transparent resolution (Figure 3a). Although the hemolysis ratio at 7.5 mg mL^−1^ reached 4.73 ± 0.45%, it is still below 5%. When 10 mg mL^−1^ diatom frustules were used to treat RBCs, a 5.64 ± 0.39% hemolysis ratio was present, which is significantly higher than the concentrations in 1, 2.5, and 5 mg mL^−1^ (*p* < 0.001). However, the data are still favorable compared to those of non-functionalized *Coscinodiscus* sp. diatom frustules (17.76 ± 1.16% at 10 mg mL^−1^) and diatomite (13.83 ± 0.11% at 5 mg mL^−1^) [17]. The obtained result denotes favorable hemocompatibility of *Navicula australoshetlandica* sp. diatom frustules. It was reported that the hemolysis activity has highly correlate to the specific chemical composition of the frustules surface functional groups [13]. The more silanol groups contact with the erythrocyte, the higher the hemolysis ratio [25]. Thus, we hypothesized that the better biocompatibility shown by *Navicula australoshetlandica* sp. may attributed to the lower quantity of silanol groups brought by the different amount of hydrogen peroxide used in purified process. Moreover, the porous network distribution and its pore diameter may be another important factor related to the hemolysis ratio. Navicula diatom, in general, possess larger pore diameter than the coscinodiscus and diatomite. Larger nanopores were able to reduce the density of –OH, resulting in lower hemolytic activity [26]. Impacts of surface functionality on diatom frustules hemolytic activity still require further exploration.

Additionally, the SEM images vividly pictured RBCs gathered around diatom frustules while still retaining their normal morphology without leaking after the hemolysis test (Figure 3c). This suggests the hemocompatibility and hemostatic potential of *Navicula australoshetlandica* sp. diatom frustules.

### 3.3. In Vitro Blood Clotting Evaluation

In vitro whole blood clotting time was introduced to measure the hemostasis effect of diatom frustules. The blood used in this evaluation was recalcified by 1/50 V 0.025 mol/L CaCl_2_. The purpose of calcium chloride here is to provide proper calcium ion (clotting factor IV) into anticoagulant blood to activate the blood coagulant [27]. In order to compare hemostatic efficiency, the blood clotting evaluation was carried out among frustules and chitosan suspended in various volumes of blood (Figure 4). Chitosan, a natural polysaccharide, has been be extensively used as commercial hemostatic material for its effective hemostatic performance, biodegradability, and biocompatibility [28]. Thus, chitosan had been widely used as an effective hemorrhage control powder. In this study, we use chitosan as the positive control. According to Figure 4a, both diatom frustules and chitosan showed significantly shorter clotting time than that of the blank control (429.37 ± 12.14 s). The diatom frustules group exhibited the shortest clotting time (134.99 ± 7.00 s) for a hemostasis material/blood (mg/μL) ratio of 1:100, which is 1.83 times (112.32 s) shorter than that of chitosan and 3.18 times (294.38 s) shorter than that of the control. When the blood volume increased from 25 to 125 μL, the clotting time of chitosan was significantly extended from 181.06 ± 7.25 to 282.52 ± 3.48 s, while that of the diatom frustule remained almost the same, changing from 167.37 ± 13.33 to 181.73 ± 7.60 s. This indicates that blood volume slightly influenced the hemostasis characteristics of diatom frustules, showing great potential to be utilized as a medical hemostasis material.

The characterization of the activated partial thromboplastin time (aPTT) and prothrombin time (PT) reveals the role the diatom plays during the hemostasis pathway. The aPTT of a diatom frustule was 44.53 ± 7.78 s shorter than the control, while offering no significant change (*p* < 0.5, analyzed using SPSS) to the PT (8.73 ± 0.12 s for blood; 8.85 ± 0.37 s for diatom frustule), as shown in Figure 4b. The shortening of aPTT implies that the frustules can induce the intrinsic pathway of blood coagulation. Previous studies have indicated that the highly porous and negatively charged surfaces of silica can activate the intrinsic pathway of blood coagulation by stimulating the coagulation factors XI and XII [29,30]. An abundant negatively charged group can be attributed to aPTT shortening, while a nearly constant PT is an indicator that the diatom frustules did not activate the extrinsic coagulation pathway.

## 4. Discussion

### 4.1. In Vitro Clotting Times of Different Diatom Frustules

Before biosilica was obtained from diatom, there were considerable experiments on hemorrhage control carried out by using mesoporous silica particles because of their desirable biodegradability and biocompatibility [4,31]. However, in the advanced synthesis technology of highly ordered mesoporous silica materials, the fabrication procedure is expensive, complicated, and is highly energy-consuming. Diatom frustule is a promising hemostatic material due to its mesoporous structure and the negative charge brought by active functional groups. It is reported that the mesoporous biosilica could rapidly promote hemostasis process by aggregating red blood cells and platelets and activating the coagulation cascade [32]. References presented that a negative charge is crucial for activating the intrinsic pathway of the coagulation cascade [33]. Due to the active reaction site of the functional groups, such as silanol groups, the relative enzyme blood coagulation factor XII in the blood serum could be the activated favorite for the blood clotting [18]. Several diatom frustules from Centricae (central symmetry) and Pennatae (radial symmetry) were reported for the hemorrhage application. The hemostatic performance of three species of centric diatoms was assessed, namely that of *Thalassiosira weissflogii*, *Thalassiosira* sp., and *Cyclotella cryptica*. *Thalassiosira* sp. showed a hemostasis time under 167.33 ± 14.74 s and *Cyclotella cryptica* showed a time of 235.67 ± 20.84 s [18]. The hemostasis time of *Coscinodiscus* sp. was 203.67 ± 15.63 s [34]. Meanwhile, the vitro coagulation time of *Pennatae* is generally shorter than that of *Centricae* frustules. It was reported that the vitro coagulation time of *Pennatae* diatom frustule such as *Navicula* sp., and *Pleurosigma indicum* were 126.66 ± 5.8 s and 205.33 ± 5.77 s, respectively [11]. In our study, the *Navicula australoshetlandica* sp. diatom frustules were investigated for the first time for the hematischesis characteristics, with the vitro coagulation time of 134.99 ± 7.00 s at a concentration of 10 mg L^−1^. The results are consistent with previously reported results on the vial inverting test of diatom frustules and synthetic silica [14]. Previous study on silica demonstrated that the accessibility and diffusion of clotting were mainly dependent on the pore size of the silica nanoparticle [35]. Thus, we hypothesized that the better performance for *Navicula australoshetlandica* sp. was brought by the highly porous surface with proper pore size on the frustules. Diatom *Navicula australoshetlandica* sp. frustule obtained an impressivly large surface area 401.45 m^2^ g^−1^ that is 8.26 times of *Navicula* sp. [36], 7.56 times of *Thalassiosira pseudonana* [37], and nearly 3 times of *Thalassosira weissflogii* [38], which shows the great possibility of its use not only as hemostasis material but also as a drug delivery system. Furthermore, *Navicula australoshetlandica* sp. was screened from aquaculture wastewater, which provided a sustainable pathway to obtain the highly-porous silica material with relatively low cost.

### 4.2. Surface Modification of Diatom Frustules Benefits Hemostatic Performance

In order to improve hemostatic performance, various modifications were introduced to diatom frustules [39,40,41]. A successful attempt was achieved in the combination of diatom frustules and clotting factors in coagulation pathway. In 2016, Li and Han et al. coated the frustules with chitosan. The shortest coagulation time of chitosan-coated frustules was 248 ± 32.42 s at 10 mg mL^−1^, 250 s shorter than that of bare diatom at the same concentration [17]. In 2018, Li et al. prepared the hybridization of diatom frustules with calcium, an important composition to coagulate factor IV. The clotting time obtained from Ca-frustules was 145.01 ± 20.41 s at 10 mg mL^−1^, which was nearly 250 s shorter than that of the frustules group at 10 mg mL^−1^ [34]. An effective promotion strategy and various modifications are the breakthrough points to promote blood clotting and hemorrhage control. In 2021, Mu et al. immobilized thrombin on the surface of polydopamine-coated diatom biosilica. This composite hemostatic material exhibited an effective clotting time with a low concentration (66 s at 5 mg mL^−1^), which was approximately 8% of the clotting time of the control group [42].

In addition to the combination, there is still much room for the diatom frustules to develop on the biomedical material. Especially, the silanol groups on the surface of diatom frustules. Although it may induce hemolytic activity, proper chemical surface modification can shield the exposure of -OH and can provide an anchor for desired functional groups, proteins, and molecules. The amino group is commonly used for the modification on the surface of silica-based materials [43]. It endows frustules with the affinity for proteins and thromboplastic drugs. In 2012, Bariana et al. successfully grafted amine groups on the diatom surface by the interaction between organosilanes and the -OH, which provided better loading properties for hydrophobic drugs such as indomethasin [44]. Genetic modification can be a powerful tool that can be used in designing frustule surface characteristics. In 2015, Delalat et al. genetically engineered *Thalassiosira*
*pseudonana* to express an IgG-binding domain of protein G on frustule surfaces. This enabled the tailoring of cell-targeting antibodies [8]. In 2020, Kumari et al. provided a method for designing diatom biosilica properties. They successfully immobilized glucose oxidase and horseradish peroxidase into the rigid part of frustules and tripled the catalytic activity than soluble enzyme [45]. In the future, we can take advantage of genetic engineering to immobilize coagulation factors in the frustule surfaces. Above all, diatom frustules can be a promising material applied in biomedical applicants with tremendous prospects.

## 5. Conclusions

In this study, we observed, for the first time, the special ordered “porous web” (6–8 nm) substructure and impressive surface area (401.45 m^2^ g^−1^) of diatom *Navicula australoshetlandica* sp. L1 frustules using microscopy and structural characterization. The low hemolysis ratio (<1.55 ± 0.06%) of these frustules indicates their great hemocompatibility. The abundant OH functional groups found on the frustule surface might contribute its short clotting time (134.99 ± 7.00 s) compared with that of chitosan. The results show that *Navicula australoshetlandica* sp. is a promising sustainable source of frustules that can be cultivated with aquaculture wastewater for the production of hemostasis material for potential use in medical applications.

## Figures and Tables

**Figure 1 materials-14-03752-f001:**
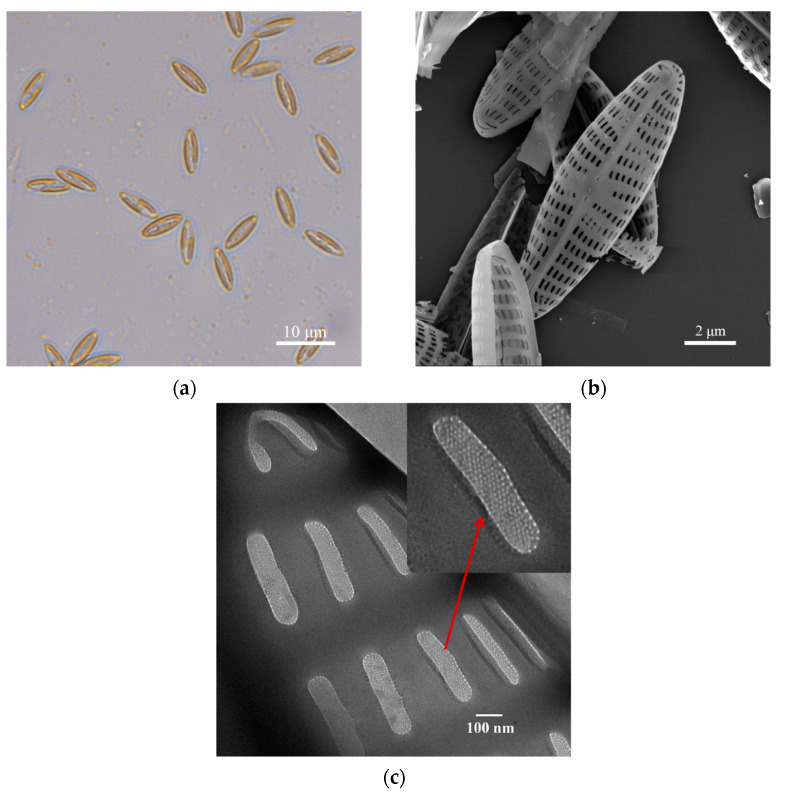
Images of *Navicula australoshetlandica* sp. living cells and frustules: (**a**) the morphology of diatom cells cultivated in f/2 medium-enriched seawater sodium observed using light microscopy; (**b**) an SEM image of *Navicula australoshetlandica* sp. frustule; (**c**) details of the frustule structure showing the different arrangement of the porous surface pictured using transmission electron microscopy. Scale bars showing (**a**) 10 μm, (**b**) 2 μm, and (**c**) 100 nm, as indicated.

**Figure 2 materials-14-03752-f002:**
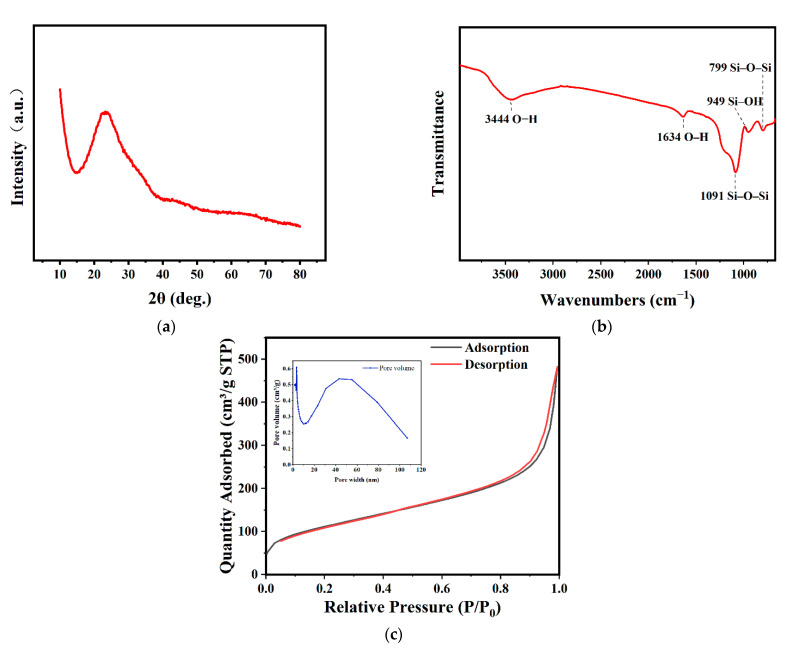
Results of physico-chemical analysis of *Navicula australoshetlandica* sp. diatom frustules: (**a**) Fourier transform infrared (FTIR) spectrum; (**b**) X-ray diffraction (XRD); and (**c**) nitrogen adsorption/desorption isotherms (77.4 K) and pore width distribution, calculated from the adsorption branch by applying BJH method.

**Figure 3 materials-14-03752-f003:**
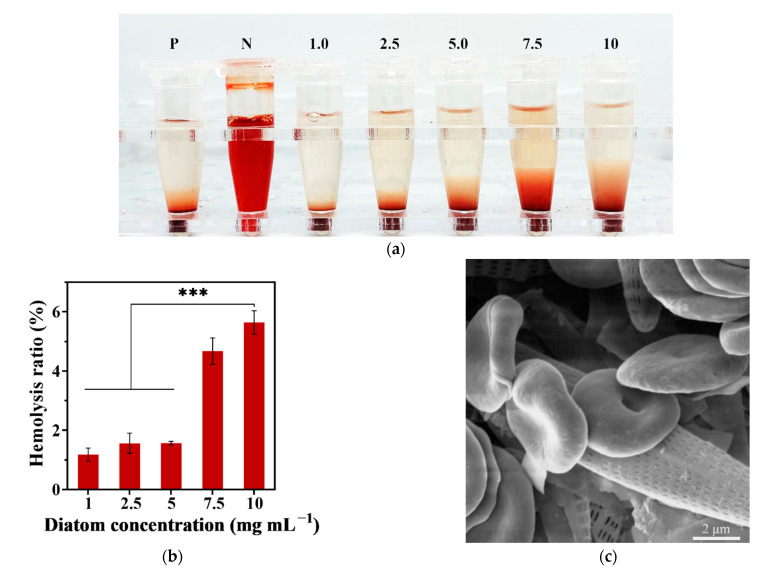
(**a**) Photographs of red blood cells (RBCs) treated with different concentrations of diatom frustules (mg mL^−^^1^); P refers to positive control, N refers to negative control. (**b**) Hemolysis ratio of different concentrations of diatom frustules in saline suspension. (**c**) Scanning electronic microscopy (SEM) image of red blood cells after hemolysis test with 2.5 mg mL^−^^1^ diatom frustules. Data represent the mean ± SD (n = 3). *** represents a significant difference, *p* < 0.001.

**Figure 4 materials-14-03752-f004:**
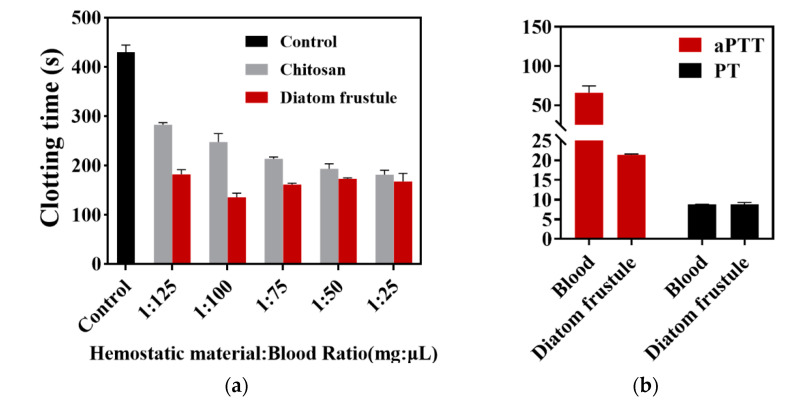
(**a**) In vitro clotting time of control (1 mL blood without additive), chitosan, and diatom frustules. Hemostatic materials were added to blood at ratios of 1:125, 1:100, 1:75, 1:50, and 1:25. Data represent the mean ± SD (n = 3). (**b**) The coagulation tests with aPTT and PT of diatom frustules and blood. Data represent the mean ± SD (n = 6).

**Table 1 materials-14-03752-t001:** The BET setting conditions.

Analysis Adsorptive	Analysis Bath Temperature	Sample Mass	Warm Free Space	Cold Free Space	Equilibration Interval
N_2_	77.30 K	0.0574 g	17.38 cm³	49.99 cm³	30 s

**Table 2 materials-14-03752-t002:** Pore structure parameters of diatom frustules.

S_BET_ (m² g^−1^)	V (cm^3^ g^−1^)	W_BJH_ (nm)	W_d_ (nm)	S_BJH_ (m^2^ g^−1^)
401.45	0.46	8.58	4.61	329.78

S_BET_ represents the BET surface area, single point adsorption total pore volume (V), the BJH pore diameter (W_BJH_), and adsorption average pore width (Wd) (4 V/S_BET_).

## Data Availability

The data presented in this study are openly availablein [all data in article Study on the Hemostasis Characteristics of Biomaterial, FigShare] at [10.6084/m9.figshare.14899092].

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
