# Peer review of "Study on the Hemostasis Characteristics of Biomaterial Frustules Obtained from Diatom Navicula australoshetlandica sp."

_materials, 2021, doi:10.3390/ma14133752_

Round 1

Reviewer 1 Report

The authors present an innovative work about investigations of haemophilic properties of biosilica from diatoms. The paper is well presented. However, some general comments need to be addressed. First, the authors claim they are the first demonstrating a presence of a nanonetwork in silica shells of this Navicula australoshetlandica subspecie. This referee does not agree with authors. A plethora of nanonetworks have been already investigated and published in 2011 (Revision of the genus Navicula s.s. (Bacillariophyceae) in inland waters of the Sub-Antarctic and Antarctic with the description of five new species; Phycologia 50(3):281-297 (2011); DOI: 10.2216/10-49.1). This sentence has been reported in the abstract. In the introduction section, the authors should define in which literature reference the statement “However, the mesoporous silica made by chemical synthesis can cause hemolytic  damage due to toxic chemical groups on its surface” has been reported (2nd page, line 55). Please correct the sentence “The diatom frustule is the silicon scaffold of a diatom cell without organic residues” in “The diatom frustule is the silica scaffold of a diatom cell without organic residues” (3rd page, line 90). This referee advices the authors to better explain how haemolysis percentages have been obtained, also in results and discussion section, and they should give a possible explanation on why coscinodiscus and diatomite shells worst perform than Navicula shells.  (around line 155-158, 6th page). In addition, why pennate diatoms should work better than round ones? In lines 168-171 the authors should give more detailed explanations of the role of calcium chloride in clotting experiments, and explain better why they used chitosan as references. Moreover, is it possible to have data about a composite combination of diatoms frustules and chitosan in clotting experiments?

As a general comment, how could the authors justify the use of materials obtained from raw aquaculture wastewater for biomedical applications? This referee asks the authors to better organize the discussion proposed in 239-251 lines (page 6), and to integrate it with general and convincing comments on the use of specific Navicula shells instead of other pennate diatoms. Please explain better the perspectival sentence “In the future, we can take advantage of this method to immobilize desired enzymes onto frustule surfaces”.

This referee expects from authors also a comment about the real application of this material and its integration into a biomedical device (e.g. in skin patch, as stabilized layer, in polymer composites?), and a more complete literature presentation about biosilica biocompatibility is needed.

Moreover, to general enrich the paper this referee suggests the author to supply information or simple data about biocompatibility of Navicula frustules at biological interface with other cells (e.g. fibroblasts, dermal cells). For all these reasons, this referee accept the publication of the paper with major revisions.

Author Response

We would like to thank the reviewer 1 for the careful and thorough reading of this manuscript and for the thoughtful comments and constructive suggestions, which help to improve the quality of this manuscript. Based on your comment and request, we have made extensive modification on the original manuscript. Our detailed response to comments follows. A revised manuscript with the correction sections red marked was attached as the supplemental material and for easy check.   The revised part is highlighted in yellow in the article.

(Note: According to the Reviewer 2’s comment, the article structure has been adjusted with the section order as below: Introduction, Materials and methods, Results, Discussion, Conclusion)

Point 1:  First, the authors claim they are the first demonstrating a presence of a nano network in silica shells of this Navicula australoshetlandica subspecie. This referee does not agree with authors. A plethora of nanonetworks have been already investigated and published in 2011 (Revision of the genus Navicula s.s. (Bacillariophyceae) in inland waters of the Sub-Antarctic and Antarctic with the description of five new species; Phycologia 50(3):281-297 (2011); DOI: 10.2216/10-49.1). This sentence has been reported in the abstract. 

Response 1: Thank you for the valuable suggestions.

Through careful investigation and comparison of the literature list above (DOI: 10.2216/10-49.1), the definition for the nano network in the manuscript was not so appropriate. The nano network claimed in our manuscript refers particularly to the pore shorter than 10 nm array into the “regular nanonetwork” (Figure 1c).  

We have been cautiously looked up the paper including SEM pictures of Navicula diatom, description similar to nano networks were reported in reference 1 and reference 2 [1,2]. However, the published paper mainly focused on Pennate diatom hemostatic research which only discussed about the size of diatom, without further investigation of its smaller pore into the highly ordered pattern of pores[3].

Above all, for the scientific strictness, the relevant part has been changed as “A special “porous web” (6–8 nm) substructure in the ordered nanopores (165–350 nm) of boat-shaped diatom frustule was observed in Navicula australoshetlandica sp. using SEM and TEM analysis.” In line 20-22.

Point 2:  In the introduction section, the authors should define in which literature reference the statement “However, the mesoporous silica made by chemical synthesis can cause hemolytic damage due to toxic chemical groups on its surface” has been reported (2nd page, line 55).

Response 2: Thanks for the reviewer’s comment. We have added the literature reference in the relevant part shown as below

Page 2 Line 54-56

“However, the synthesis of mesoporous silica requires multiple agents, co-solvents, or additives as well as the finer control of the synthesis process to create the desired shape and size [14,15].”

Point 3:  Please correct the sentence “The diatom frustule is the silicon scaffold of a diatom cell without organic residues” in “The diatom frustule is the silica scaffold of a diatom cell without organic residues” (3rd page, line 90).

Response 3: Thank you for your valuable comments and suggestions on our manuscript,we have modified the manuscript accordingly. The relevant part has been revised in the manuscript shown as below:

Page 6 Line 208-209

“The diatom frustule is the silica scaffold of a diatom cell without organic residues.”

Point 4:  This referee advices the authors to better explain how hemolysis percentages have been obtained, also in results and discussion section, and they should give a possible explanation on why coscinodiscus and diatomite shells worst perform than Navicula shells.  (Around line 155-158, 6th page). In addition, why pennate diatoms should work better than round ones?

Response 4: Thank you for your valuable comments and suggestions on our manuscript. We have modified the manuscript accordingly. The relevant part has been revised in the manuscript shown as below:

Page 8 Line 26-269

” To quantify the hemolytic effects of diatom frustules in SD rats erythrocytes, hemolysis ratio was calculated (according to equation 1.).”

Page 8 Line 279-289

“It was reported that the hemolysis activity has highly correlate to the specific chemical composition of the frustules surface functional groups[4]. The more silanol groups contact with the erythrocyte, the higher the hemolysis ratio[5]. Thus, we hypothesized that the better biocompatibility shown by Navicula australoshetlandica sp. may attributed to the lower quantity of silanol groups bring by the different amount of hydrogen peroxide used in purified process. Besides, the porous network distribution and its pore diameter may be another important factor related to the hemolysis ratio. Navicula diatom in general possess larger pore diameter than the coscinodiscus and diatomite. Larger nanopores was able to reduce the density of -OH, leading to lower hemolytic activity[6]. Impacts of surface functionality on diatom frustules hemolytic activity still need further exploration.”

Page 10 Line 354-357

“Previously study on silica demonstrated that the accessibility and diffusion of clotting were mainly dependent on the pore size of the silica nanoparticle [34]. Thus, we hypothesized that the better performance for Navicula australoshetlandica sp. was bring by the highly porous surface with proper pore size on the frustules.”

Point 5:  In lines 168-171 the authors should give more detailed explanations of the role of calcium chloride in clotting experiments, and explain better why they used chitosan as references.

Response 5: Thanks for the reviewer’s comment. Calcium as crucial composition of the coagulation trigger agent is participating in activating both intrinsic and extrinsic pathway of blood coagulant[7,8]. Calcium chloride here is used to provide proper calcium ion (clotting factor â…Ł) into anticoagulant blood to reactive the blood coagulant. Chitosan, a natural polysaccharide, have been be extensively used as commercial hemostatic material for its effective hemostatic performance, biodegradability, and biocompatibility. Also, it can be sustainable obtained in a relatively cheap powder hemostatic material. Diatom biosilica frustules as a new material applied in hemorrhage control are of nice biocompatibility and can get metabolized and cleared from the body. With large specific surface area but low density, diatom frustules present a form of lightweight powder. Moreover, chitosan is widely used in clinical applications and definite hemostatic effect. Above all, we believe that chitosan could be a suitable reference in blood clotting test.

The detailed explanations of the role of calcium chloride in clotting experiments, and the reason why chitosan was used as references have been added in the manuscript as below:

Page 9 Line 302-303

“The calcium chloride here is to provide proper calcium ion (clotting factor â…Ł) into anticoagulant blood to reactive the blood coagulant.[7]”

Page 9 Line 305-308

“Chitosan, a natural polysaccharide, have been be extensively used as commercial hemostatic material for its effective hemostatic performance, biodegradability, and biocompatibility[9]. Thus, chitosan had been widely used as an effective hemorrhage control powder. In this study, we use chitosan as positive control.”

Point 6:  Moreover, is it possible to have data about a composite combination of diatoms frustules and chitosan in clotting experiments?

Response 6: Thanks for the reviewer’s comment. It is possible to combine diatom frustules with chitosan. However, the chitosan is a kind of polysaccharide that have poor water-soluble properties, needed to be dissolved in alkaline solution. As a result, the physically coating frustules with chitosan will bring the material into an alkaline environment. Besides, there is a worry that the chitosan may block the pores in diatom frustules, which may shorten the advantage of diatom frustules.

Different preparation technologies have been applied to fabricate chitosan hemorrhage materials in hydrogel, membrane, sponge etc. forms. The preparation process including solvent evaporation method, In situ deposition method, freeze-drying method etc. The process requires complex crafts. 

The solvent evaporation is a method to achieve crystal growth by increasing the concentration of solute through the evaporation of solvents to obtain a saturated solution. To obtain the chitosan membrane with controllable microporous structures, Zeng et al. added silicon particles with different diameters into acetic acid dissolved chitosan solution. After solvent evaporation, the silicon particles could be inserted in the chitosan membrane. Then, the membrane was immersed in sodium hydroxide, leading the dissolution of silicon particles and the formation of the chitosan membranes with micropores.

The electrochemical deposition method refers to the technology of forming a coating in a matrix under an external electric field, in which the negative ions migrate in the electrolyte solution, and the redox reaction of electrons occurs in the electrode. Based on the principle that the chitosan in a weak acidic condition presents a positive charge and not charged in alkaline states, chitosan always is deposited in the cathode by the electrochemical deposition method.   

Before, chitosan sponge has been successfully combined with diatom frustules, the clotting time was successfully shortened for nearly 250 s[10]. A freeze-drying method uses the sublimation principle to dry the freezing material. First, the dried stuff is frozen at low temperatures, and then the frozen water molecules directly sublimate into the vacuum environment by adjusting the temperature. After freeze-drying, the material has a porous sponge shape. Porous sponge combine porous frustules provides a promising prospect. But it needs additional experiments on the chitosan materials formulation.

In this assay, we focused on the fundamental experiments of a new obtained diatom frustules’ chemical and physical properties of and on the exploration of the potential to be applied in hemostatic materials. The purposes can be realized in existing assay’s design. Thanks again for the reviewer’s comments. We will consider adding this content in our subsequent experiments.

Point 7:  As a general comment, how could the authors justify the use of materials obtained from raw aquaculture wastewater for biomedical applications?

Response 7: Thanks for the reviewer’s comment. The safety and biocompatibility of the material is a precondition to applied them in biomedical applications.

The water used in aquaculture itself depends that the composition of the water is not harmful to the growth of organisms. Wastewater discharged from aquaculture contains high concentrations nitrogenous compounds (ammonia, nitrite, and nitrate), phosphorus and dissolved organic carbon[11]. These compounds have caused major environmental impacts. However, algae like diatom can assimilate them as a source of nitrogen rather than enriched the toxic effluents[12]. Diatoms grown on fish farm effluents (enriched in nitrogen and phosphorus) may be used as feed for bivalves[13].

In this assay, we do not directly utilize the raw diatom obtained from the aquaculture wastewater, but use the diatom scaffold after removing the organic substance through acid solution digestion. After digestion, the amount of organic material in and on the surface of the frustules had significantly reduced[14].

Besides, diatomaceous earth (diatomite), the is a fossil material of sedimentary origin, formed over centuries by diatom siliceous skeleton. They consist of 70–90 % silica, clay, and a few metallic oxides, such as Al2O3 and Fe2O3[15]. There are abundant cases in different biomedical applications as drug carrier, like the carrier of siRNA to transport inside human cancer cells, deliver water-soluble anticancer drug, and Extend drug release time[16-19]. Above all, we believe that the diatom cultivated from the aquaculture wastewater is a promising way to obtain biosilica for the commercial application.

Point 8: This referee asks the authors to better organize the discussion proposed in 239-251 lines (page 6), and to integrate it with general and convincing comments on the use of specific Navicula shells instead of other pennate diatoms. Please explain better the perspectival sentence “In the future, we can take advantage of this method to immobilize desired enzymes onto frustule surfaces”.

Response 8: Thanks for the reviewer’s comment. The discussion had been reorganized as below. In terms of the general and convincing comments on the use of specific Navicula shells. For now, the application of diatom frustules on hemorrhage control is still in its infancy, even the application in drug delivery[20]. The papers related to diatom frustules hemorrhage control, whether it is Navicula shells or the shells from other diatom, had been exhaustivity cited in this paper[3]. Especially, “Tentative identification of key factors determining the hemostatic efficiency of diatom frustule” (DOI: 10.1039/d0bm02002h) is the only paper which used pennate diatom including Navicula sp. into the study. So as to provide a direction to realize the full potential of diatom frustules on hemorrhage control, we revised the relevant part as below.

Page 10 Line 367-368

“Successful attempt was achieved in the combination of diatom frustules and clotting factors in coagulation pathway.”

Page 10 Line 379-396

“In addition to the combination, there is still much room for the diatom frustules to develop on the biomedical material. Especially, the silanol groups on the surface of diatom frustules. Though it may induce hemolytic activity, proper chemical surface modification can shield the exposure of -OH and can provide an anchor for desired functional groups, proteins, molecules. The amino group, commonly used for the modification on the surface of silica-based materials [42]. It endows frustules with affinity for proteins and thromboplastic drugs. In 2012, Bariana et al. successfully grafted amine groups on the diatom surface by the interaction between organosilanes and the -OH, that provided better loading properties for hydrophobic drugs such as indomethasin [43]. Genetic modification can be a powerful tool that can be used in designing frustule surface characteristics. In 2015, Delalat et al. genetically engineered Thalassiosira pseudonana to express an IgG-binding domain of protein G on frustule surfaces. This enabled the tailoring of cell-targeting antibodies [8]. In 2020, Kumari et al. provided a method for designing diatom biosilica properties. They successfully immobilized glucose oxidase and horseradish peroxidase into the rigid part of frustules and tripled the catalytic activity than soluble enzyme [44]. In the future, we can take advantage of genetic engineering to immobilize coagulation factors in the frustule surfaces. Above all, diatom frustules can be a promising material applied in biomedical applicants with tremendous prospects.”

Point 9: This referee expects from authors also a comment about the real application of this material and its integration into a biomedical device (e.g. in skin patch, as stabilized layer, in polymer composites?), and a more complete literature presentation about biosilica biocompatibility is needed. Moreover, to general enrich the paper this referee suggests the author to supply information or simple data about biocompatibility of Navicula frustules at biological interface with other cells (e.g. fibroblasts, dermal cells).  

Response 9: Thank you for your useful comments and suggestions on our manuscript,we have modified the manuscript accordingly.

Real applications and biodemical devices of biosilica from diatom have already made into composite beads, sponge, aerogel etc.The composite beads for hemostasis were prepared by alkalization precipitation method by using chitosan, dopamine, and diatom Coscinodiscus sp.. The porous internal structure of CDDs led to rapid and large amount of water absorption, which contributed to the rapid hemostasis (83 s, 22% of the control group). The hemolytic rate of CDDs was less than 5% and fibroblast L929 cell viability was above 80%, confirming its good biocompatibility[21].

In 2020, Li J et al study, a chitosan/diatom-biosilica-based aerogel is developed using dopamine as cross-linker by simple alkaline precipitation and tert-butyl alcohol replacement. 30% tert-butyl alcohol replacement of aerogel possesses the largest surface area (74.441 m2 g−1), water absorption capacity (316.83 ± 2.04%), and excellent hemostatic performance in vitro blood coagulation (≈ 70 s). Furthermore, this aerogel exhibits the shortest clotting time and lowest blood loss in rat hemorrhage model. Besides, different concentrations of aerogel (10, 5, 2.5, 1.25, and 0.625 mg mL−1) was incubated with L929 cells to evaluate the cytotoxic effects in vitro. The cell viability at all tested concentrations was remained at high level (higher than 94%) when the incubation time was extended from 24 to 72 h[22].

In 2021, Sun et al. combined alkylated chitosan (AC) and diatom biosilica to develop a safe and effective hemostatic composite sponge (AC-DB sponge) for hemorrhage control. It exhibited rapid hemostatic ability in vitro (clotting time was shortened by 78% than that of control group), with favorable biocompatibility (hemolysis ratio < 5%, no cytotoxicity). Meanwhile, AC-DB sponge had excellent performance in in vivo assessments with shortest clotting time (106.2 s) and minimal blood loss (328.5 mg). All above results proved that AC-DB sponge had great potential to be a safe and rapid hemostatic material[23].Also, application in skin-attachable chitosan-diatom triboelectric nanogenerator[24], cell-growth and bone-mineralisation platforms[25].

In future, the applicants will get developed for more biomedical device. Biosilica biocompatibility have been widely proved in cellular and rat levels. human Saos-2 osteoblastic cell line (ICLC) and NHDF fibroblasts (PromoCell) were used for cell culture experiments. In 2015, Delalat et al. conducted in vivo biodistribution studies of biosilica and assessed whether the biosilica will caused the tissue damage. After a single intravenous injection of biosilica obtained from T. pseudonana into nude mice,

mice, animals were observed daily for 8 days and the major organs were then collected. During observation, none of the mice exhibited anyobservable symptoms of acute tissue damage. Optical microscopy studies on tissue sections (8 mm) did not reveal any noticeable abnormality in the major organs, the brain, heart, kidney, liver and lung, or the tail. Degraded diatom biosilica was observed in liver and kidney sections but not in the lung[26]. In 2016, Cicco et al. conducted viability assay of bare frustules from Thalassiosira weissflogii on the NHDF fibroblasts and Saos-2 osteoblasts cells. After 96h incubation, a statistically significant cell growth (96h vs. 24h, p < 0.01) is presents for both cells[27].

Besides, diatomite nanoparticles (DNPs) had also been proved the biocompatibility in cancer cells. In 2017, Managò et al. incubated the DNPs with human lung epidermoid carcinoma cell line (H1355) up to 72 hours are investigated by Raman imaging. A considerable DNPs uptake in cells is observed within 6 hours, with equilibrium being achieved after 18 hours. The obtained data show the presence of DNPs up to 72 hours, without damage to cell viability or morphology[28].

Reviewer 2 Report

Comments on "Study on the hemostasis characteristics of biomaterial frustules obtained from diatom Navicula australoshetlandica sp."
The manuscript is interesting and easy to read. Before publication, some points deserve attention:
1- line 29- aPPT is maybe aPTT? Furthermore, please add the fulll name before using the abbreviation.
2- line 89- cited Section 2.1 is actually section 4.1.
3- line 99 -please correct figure 2b into 2a.
4- Figures 1a and 1b are never cited in the text.
5- in the caption of figure 2, the meaning of "substructure" is not appropriate:  I suggest to replace it with "physico-chemical"
6- Table 1 is never cited in the text
7- line 148-please add "red blood cells" and put RBCs into parentheses
8- in figure 3: b and c have been exchanged. Furthermore figure 3b is never cited in the text.
9- For clarity of presentation, I suggest to move section Materials and Methods immediately after Introduction and before Results.
10- line 271- table 1 does not contain any details about experiments, but BET results. Please add a Table here with experimental information.
11- line 278- please insert correct number of cited section
12- lines 303-304- please declare the amount of material used for BET analyses.
13- line 326- Anima should be changed into Animal

Author Response

Responses to Reviewer 2 Comments

We would like to thank the Reviewer 2 for the careful and thorough reading of this manuscript and for the thoughtful comments and constructive suggestions, which help to improve the quality of this manuscript. Our detailed response to comments follows. The revised part is highlighted in yellow in the article.

Point 1: line 29- aPPT is maybe aPTT? Furthermore, please add the full name before using the abbreviation.

Response 1: Thanks for the reviewer’s comment. The relevant part has been revised in the manuscript shown as below:

Page 1 Line 29

“The activated partial thromboplastin time (aPTT) of diatom frustule was also 44.53 s shorter than the control.”

Point 2: line 89- cited Section 2.1 is actually section 4.1.

Response 2: Thanks for the reviewer’s comment. We have removed all typos. The relevant part has been revised in the manuscript shown as below:

Page 6 Line 207-208

“The symmetric Navicula australoshetlandica sp. diatom was isolated from aquaculture wastewater and cultivated in the conditions listed in Section 2.1.”

Point 3:  line 99 -please correct figure 2b into 2a.     

Response 3: Thanks for the reviewer’s comment. We have removed all typos. The relevant part has been revised in the manuscript shown as below:

Page 7 Line 227-228

“The XRD patterns of Navicula australoshetlandica sp. diatom frustules are shown in Figure 2a.”

Point 4:  Figures 1a and 1b are never cited in the text.

Response 4: Thanks for the reviewer’s comment. We had added the citation of figure1a and 1b into the manuscript. The relevant part has been revised in the manuscript shown as below: 

Page 6 Line 211-212

“Figure 1a shows the original shape and the color of the Navicula australoshetlandica sp.. The apical axes of diatom cells varied from 8 to 11 μm, with the diameter of the highly ordered fracture porous pattern ranging 165–350 nm (Figure 1b).”

Point 5: in the caption of figure 2, the meaning of "substructure" is not appropriate:  I suggest to replace it with "physico-chemical" 

Response 5: Thanks for the reviewer’s comment. The relevant part has been revised in the manuscript shown as below:

Page 6 Line 224-226

“Figure 2. Results of physico-chemical analysis of Navicula australoshetlandica sp. diatom frustules: (a) Fourier transform infrared (FTIR) spectrum; (b) X-ray diffraction (XRD); and (c) nitrogen adsorption/desorption isotherms (77.4 K) and pore width distribution, calculated from the adsorption branch by applying BJH method.”

Point 6: Table 1 is never cited in the text

Response 6: Thanks for the reviewer’s comment. We had added the citation of original Table 1 (now Table 2) into the manuscript. The relevant part has been revised in the manuscript shown as below:

Page 7 Line 244-245

“The Brunauer–Emmett–Teller (BET) method was applied in the analysis of the specific surface area (Table 1)”

Point 7: line 148-please add "red blood cells" and put RBCs into parentheses

Response 7: Thanks for the reviewer’s comment. The relevant part has been revised in the manuscript shown as below:

Page 8 Line 267

The hemolysis ratio of red blood cells (RBCs) basically turned higher when the concentration of diatom frustules was increased, as shown in Figure 3a.

Point 8: in figure 3: b and c have been exchanged. Furthermore, figure 3b is never cited in the text.

Response 8: Thanks for the reviewer’s comment. The picture arrangement of Figure 3 has been exchanged. We had added the citation of figure 3b (now is figure 3c) into the manuscript. The relevant part has been revised in the manuscript shown as below:

Page 9 Line 291-293

“Additionally, the SEM images vividly pictured RBCs gathered around diatom frustules while still retaining their normal morphology without leaking after the hemolysis test (Figure 3c).”

Point 9: For clarity of presentation, I suggest to move section Materials and Methods immediately after Introduction and before Results. 

Response 9: Thanks for the reviewer’s comment.

The journal provides a submission template for authors. The section arrangement was following the template. But we looked over the paper issued on Materials recently, and found that the sections are not rigidly limit. So we had moved section Materials and Methods immediately after Introduction and before Results. Also, the section 1 etc cited in assay had been accordingly revised.  

Point 10: line 271- table 1 does not contain any details about experiments, but BET results. Please add a Table here with experimental information. 

Response 10: Thanks for the reviewer’s comment. We have added a table with experimental information into the manuscript. The relevant part has been cautiously revised in the manuscript shown as below:

Page 7 Line 238-239

Table 1. The BET setting conditions.

Analysis Adsorptive

Analysis Bath Temperature

Sample Mass

Warm Free Space

Cold Free Space

Equilibration Interval

N2

77.30 K

0.0574g

17.38 cm³

49.99 cm³

30 s

Point 11: line 278- please insert correct number of cited sections. 

Response 11: Thanks for the reviewer’s comment. We have removed all typos. The relevant part has been cautiously revised in the manuscript shown as below:

Page 3 Line 105-107

“The cultivate conditions were the same as in Section 2.1 (without aeration) and were conducted in triplicate.”

Page 4 Line 178-180

” The blood cells with diatom frustules were dehydrated with graded alcohol and dried at room temperature, and were subsequently observed by SEM, according to Section 2.4.”

Page 6 Line 207-208

” The symmetric Navicula australoshetlandica sp. diatom was isolated from aquaculture wastewater and cultivated in the conditions listed in Section 2.1.”

Point 12: lines 303-304- please declare the amount of material used for BET analyses.

Response 3: Thanks for the reviewer’s comment. The relevant part has been revised in the manuscript shown as below:

Page 3 Line 131-132

“Fifty milligrams diatom frustules are needed for the BET analysis compared with the 2 mg used for SEM observation.”

Point 13: line 326- Anima should be changed into Animal

Response 3: Thanks for the reviewer’s comment. We have removed all typos. The relevant part has been revised in the manuscript shown as below:

Page 4 Line 167-169

“The experiment was approved by the Experimental Animal Ethics Committee (AEWC) of Shenzhen University with the Approval No.2021002”

Round 2

Reviewer 1 Report

This referee has carefully read the modifications. I am still not convinced by the motivation set to avoid the combination of chitosan and biosilica. Chitosan can also be solubilized in aqueous media at pH 5.5-6.5, after stirring over night. This referee has experience in that. Overall, I kindly suggest the editor to accept the publication of this paper after this point-by-point revision.

This manuscript is a resubmission of an earlier submission. The following is a list of the peer review reports and author responses from that submission.